# The Price of Success—The Long-Term Outcomes of Children with Craniopharyngioma—Two Institutions’ Experience

**DOI:** 10.3390/children10071272

**Published:** 2023-07-24

**Authors:** Aleksandra Napieralska, Marek Mandera, Ryszard Sordyl, Aleksandra Antosz, Barbara Bekman, Sławomir Blamek

**Affiliations:** 1Radiotherapy Department, MSC National Research Institute of Oncology Gliwice Branch, 44-101 Gliwice, Poland; slawomir.blamek@io.gliwice.pl; 2Department of Pediatric Neurosurgery, Medical University of Silesia, 40-752 Katowice, Poland; mmandera@sum.edu.pl (M.M.); ryszard.sordyl@wp.pl (R.S.); 3Department of Pediatrics and Pediatric Endocrinology, Faculty of Medical Science, Medical University of Silesia, 40-752 Katowice, Poland; ola_antosz@tlen.pl; 4Department of Pediatrics and Pediatric Endocrinology, Upper Silesian Medical Center in Katowice, 40-752 Katowice, Poland; 5Radiotherapy and Brachytherapy Planning Department, MSC National Research Institute of Oncology Gliwice Branch, 44-101 Gliwice, Poland

**Keywords:** craniopharyngioma, pediatric oncology, endocrinopathies, long-term complications

## Abstract

An analysis of patients below 21 years old treated due to craniopharyngioma in the years 1979–2022 was performed with the aim of evaluating the long-term outcome and treatment side-effects. The standard statistical tests were used, and 56 patients with a median age of 11 years were evaluated. Surgery was the primary treatment in 55 patients; however, in only 29 it was the only neurosurgical intervention. Eighteen children were treated with radiotherapy (RTH) in primary treatment. The most common neurosurgical side effects observed were visual and endocrine deficits and obesity, which were diagnosed in 27 (49%), 50 (91%), and 25 (52%) patients, respectively. Complications after RTH were diagnosed in 14 cases (32%). During the median follow-up of 8.4 years (range: 0.4–39.8 years), six patients died and the 5- and 10-year overall survival was 97% and 93%, respectively. Five-year progression-free survival for gross total resection, resection with adjuvant RTH, and non-radical resection alone was 83%, 68%, and 23%, respectively (*p* = 0.0006). Surgery combined with RTH provides comparable results to gross tumor resection in terms of oncologic outcome in craniopharyngioma patients. Adjuvant irradiation applied in primary or salvage treatment improves disease control. The rate of complications is high irrespective of improved surgical and radiotherapeutic management.

## 1. Introduction

Craniopharyngiomas are benign brain tumors that are usually diagnosed in children and young adults. Only a few cases in older patients were described. So far, no clear gender association was found [1,2,3,4,5,6,7,8,9,10,11,12,13,14,15]. Based on the molecular profile, two subtypes of craniopharyngioma could be diagnosed: adamantinomatous (more common in children, caused by somatic mutations in gene CTNNB1) and papillary (predominantly in adults, somatic BRAF-V600E mutations present) [3,16,17,18,19,20,21,22,23,24,25,26,27,28,29,30,31]. Primary symptoms of the disease are usually related to increased intracranial pressure, compression, or invasion of the hypothalamic–pituitary axis and optic chiasm [1,2,3]. The management of children with craniopharyngioma requires a multidisciplinary approach, including pediatric neurosurgeons, radiologists, radiation oncologists, and pediatric endocrinologist [3,4,15]. Despite improvements in diagnostics, neurosurgery, and radiotherapy techniques, there is still large number of children who suffer from treatment complications [32,33,33,34,35,36,37,38,39,40,41,42,43,44,45,46,47,48,49,50,51,52,53,54,55,56,57,58,59,60,61,62,63]. Late consequences include hypothalamic obesity, hypopituitarism, worsening of psycho-social functioning, and decline in quality of life [3,49,50,51,52,53,54,55,56,57,58,59,60,61,62,63]. The rate of neurosurgical complication was observed in many studies as high as 90%, even with improved surgical approaches [50,51,52,53,54,55,56,57,58,59,60,61,62,63,64]. There are still controversies regarding the optimum management as only two prospective trials led by the German group were completed so far [5,58]. The aim of our study is to assess the long-term outcome and treatment toxicity of children and young adults below 21 years old treated due to craniopharyngioma in the last 40 years in two hospitals.

## 2. Materials and Methods

A retrospective analysis of children with craniopharyngioma who were treated between 1979 and 2022 was performed. This study was conducted in cooperation among 3 departments: radiotherapy, pediatric neurosurgery, and pediatric endocrinology. The study included all consecutive patients younger than 21 years old diagnosed and treated due to craniopharyngioma. In all cases, the diagnosis was based on diagnostic imaging (computed tomography, CT, and, in later years of the study, magnetic resonance, MR), and in all but one case, pathologic examination of the tumor samples obtained during a biopsy or surgery. At each step of the treatment, the rate of complications was obtained, and every worsening of the symptoms was scored. The study was approved by the Ethical Committee at the MSC National Research Institute of Oncology in Gliwice (number: KB/430-55/22) and was performed according to the Helsinki Declaration.

### 2.1. Neurosurgery

In the majority of cases, surgery was performed in a cooperating Pediatric Neurosurgery Department by an experienced neurosurgeon. Among 55 patients who had surgery, one patient had a transsphenoidal removal of the tumor and all the others had various types of craniotomy, with fronto-temporo-sphenoidal craniotomy being the most common approach (in 27 cases). The others had: frontal craniotomy (15 patients), bifrontal craniotomy (2 patients), parietal craniotomy (3 patients), fronto-parietal craniotomy (3 patients), fronto-temporal craniotomy (3 patients), and fronto-orbital craniotomy (1 patient). The most important tumor-related factor affecting the extent of resection was hypothalamic involvement of the tumor. In each patient, preoperative assessment of the hypothalamic destruction was made according to the Puget scale [32]. Total radical resection (R0) was attempted only for Puget 0 or 1 tumors, but not for Puget 2. In patients with the postfix chiasm, the chiasm is typically displaced superiorly and posteriorly, which facilitates access to the tumor through the widened interoptic and subchiasmatic spaces. On the other hand, in cases with the prefix chiasm, the interoptic and subchiasmatic spaces are narrowed. Therefore, it is usually necessary to use the access through the lamina terminalis, which does not always allow obtaining the R0 resection range. A third important factor limiting the extent of resection was the presence of extensive calcifications in the tumor. They are resistant to the Cavitron Ultrasonic Surgical Aspirator, and additionally they often present strong adhesions with surrounding structures (mainly with the hypothalamus and pituitary stalk), which means that attempts to remove those tumors radically significantly increase the risk of hypothalamic complications. The neurosurgeon evaluated the extent of the resection based on intraoperative assessment and postoperative imaging as follows: R0—total radical resection, R1—subtotal resection (more than 90% of the tumor was removed based on postoperative MR assessment), R2—partial resection (less than 90% of the tumor was removed), R3—tumor biopsy, R4—cyst drainage. The number of patients who underwent R0, R1, R2, R3, and R4 resection was: 7 (13%), 31 (53%), 12 (22%), 2 (4%), and 3 (6%), respectively.

### 2.2. Radiotherapy (RTH)

To ensure treatment position reproducibility, all patients were treated with the head fixed using an individual mask system (thermoplastic mask in the majority of cases). Before the year 2000, the radiotherapy treatment planning was two-dimensional (2D). The treatment planning was based on 3D imaging in all cases treated after the year 2000. The planning CT was performed in the supine position. The target volumes and normal structures were define using CT scan slices (and MR images in the later years of the study) usually 1 to 3 mm thick, from the vertex to the bottom of the cervical spine. Based on the current guidelines, the gross tumor volume (GTV) was defined as the visible lesion on post-contrast T1- weighted and T2-weighted MRI sequences with a 1 mm slice thickness. This ensures precise contouring of the solid and cystic part [7,8.62]. Prior to MR era, GTV was considered to be contrast-enhanced tumor and tumor bed visible on preoperative and postoperative CT. GTV was expanded by an anatomically constrained margin, which ranged from 0 to 7 mm to create clinical target volume (CTV). An additional geometric expansion was added to the CTV in order to create the planning target volume (PTV). RTH was delivered with the use of 6–20 MV X photon beams in patients treated after the year 2000. Before 2000, a cobalt machine (beam energy 1.25 MeV) was used for treatment.

### 2.3. Statistical Analysis

The main study endpoints were treatment-related toxicity, patients’ overall survival (OS), and progression free survival (PFS). At each step of the treatment, the rate of complications was obtained, and every worsening of the symptoms was scored. OS was calculated from the date of tumor diagnosis to the date of death of the patient or the last follow-up visit. PFS was evaluated from the date of the last day of primary treatment (surgery or radiotherapy) to the date of progression of the tumor or death of the patient. Progression was assessed with diagnostic imaging (MRI in majority of patients). The following parameters were included into univariate analysis: age, disease symptoms at the time of diagnosis, size of the tumor, date of the primary diagnosis, number of surgeries, type of surgery (the extent of resection), type of radiotherapy (conventional versus stereotactic), total dose of RTH, time interval between the first surgery and RTH, and the date of the first disease progression and death. Statistica software was used for statistical analysis (version 12.0). The Kaplan–Meier method was used to estimate OS and PFS. Median follow-up was calculated using the Kaplan–Meier method with the reverse meaning of the status indicator. Comparisons were made using the log-rank test. For the univariate and multivariate analyses of the prognostic factors, the Cox proportional hazards model was used. Variables with a *p*-value of <0.05 in the univariate Cox analysis were used for the multivariate Cox analysis.

## 3. Results

### 3.1. Patients’ Characteristics

The study group consisted of 56 consecutive patients. In Table 1, we present the comparison of the patients’ and treatment characteristics based on the year of primary treatment applied. At the time of diagnosis, the most common symptoms were: visual deficits (46%), endocrine deficits (29%), headaches (61%), vomiting/nausea (39%), disturbances of consciousness (9%), epilepsy (4%), or other (27%). Two patients (4%) had tumor diagnosed incidentally in a brain MRI performed due to other reasons. Most of the patients had a diagnosis based on only MR (36%) or MR with CT (43%) except for those treated in the early years of the study (21%). The mean tumor dimensions at the time of diagnosis were 32 × 30 × 28 mm (ranged from 7 to 93 mm in the largest dimension)—see example in Figure 1. The most common primary symptoms were headaches (61%), visual deficits (46%), and nausea or vomiting (39%). In 16 cases, endocrine deficits were present before the tumor diagnosis, with the most common being related to a low level of growth hormone (GH) and luteinizing hormone/follicle-stimulating hormone (LH/FSH) present in 7 and 6 cases, respectively.

### 3.2. Primary Treatment

In all but one case, surgery was the first step of the treatment. The extent of the resection was documented in all the patients. R1 resections were the most commonly performed (over 50% of surgeries, regardless of the number of interventions). During the course of the disease, a shunt was placed in 15 patients (27%). Table 1 contains detailed information regarding the type of the first neurosurgical intervention. Surgery was the only primary treatment method used in the case of 38 patients.

Eighteen patients were treated with adjuvant RTH (in one case, RTH was the only treatment method applied). Conventionally fractionated RTH techniques were used in 9 patients: 2D RTH (1 patient), 3D conformal RTH (1 case), intensity modulated RTH (IMRT—5 cases), and volumetric modulated arc therapy (VMAT—2 cases) with 3 to 10 fields. Patients who received treatment with Cobalt-60 were usually treated with two opposing fields (4 patients). Five patients were treated with stereotactic RTH, and 4 to 15 fields were used. Two patients received a single fraction of 6 Gy of stereotactic boost after the end of conventional RTH (after a total dose of 45 and 52.2 Gy, respectively). The total dose used ranged from 45 to 54 Gy (median 54 Gy) in case of patients treated with conventional irradiation and from 12 to 18 Gy (median 15 Gy) in those who received stereotactic treatment. The mean volume of GTV, CTV, and PTV was 11.5 ± 8 cc, 27± 23 cc, and 43 ± 33 cc, respectively.

### 3.3. Recurrences

Disease progression was diagnosed in 31 patients after the end of primary treatment. Median PFS was 71 months (Figure 2). Patients who underwent R0 surgery alone had 5-year PFS of 83%. Equal five- and ten-year PFS of 68% were observed in patients after combined treatment (surgery with adjuvant irradiation). In the univariate analysis, the positive impact on PFS has a lack of preoperative visual deficits (*p* = 0.021) and adjuvant irradiation applied in the primary treatment (*p* = 0.001). Multivariate analysis confirmed that adjuvant RTH was significantly associated with improved PFS (*p* = 0.004, Table 2).

### 3.4. Salvage Treatment

The analysis of the type of the recurrence treatment was performed. Among all who were diagnosed with the progression of the disease, one patient did not receive treatment. All the others had local therapy applied surgery in 24 cases (combined with adjuvant irradiation in 15 cases) and 6 had RTH as the only salvage therapy method. The extent of the resection of the recurrent tumor was evaluated in 23 cases, and R0, R1, R2, and R4 surgery was performed in 2, 15, 5, and 1 case, respectively. The type of surgery did not affect the time to the next tumor progression (*p* = 0.525). Among those who received irradiation, 11 were treated with conventional RTH and 10 had stereotactic treatment (in one case combined with conventional RTH). Total dose ranged from 45 to 54 Gy (fraction dose of 1.8 Gy) in the case of conventional RTH (one had additional 5 Gy stereotactic boost), and from 12 to 30 Gy in the stereotactic group (delivered with fraction dose of 5 to 16 Gy). The best outcome was observed in patients who received combined treatment and in the RTH-only group (*p* = 0.001).

In one patient, 14 years after the primary diagnosis, a second extracerebral brain tumor was found on an MRI in the left frontal area. The patient experienced two recurrences of the primary tumor, which eventually were successfully treated with repeated surgeries (from left frontal access) and repeated irradiations. At the time of the second tumor diagnosis, the size of the primary tumor was stable. The patient was again referred for neurosurgical intervention and had gross-total resection of the left frontal area tumor. The histopathologic examination of tissue samples revealed craniopharyngioma with Ki 67 of 1%. Afterwards, with no adjuvant treatment, the patient is in regular MRI control with no signs of recurrence in the frontal area and no progression at the primary tumor site.

### 3.5. Overall Treatment Toxicity

Only three patients had no complications after the surgery. The most common neurosurgical side effects observed were visual and endocrine deficits and obesity, which were diagnosed in 27 (49%), 50 (91%), and 25 (out of 48–52%) patients. Sixteen patients had endocrine deficits observed before the neurosurgical intervention; however, in all of them, the severity of hypothalamic–pituitary axis damage was more pronounced. Other complications were scarce, including vascular complications in three patients, changes in the behavior in two patients, hemiparesis in two, and aphasia in one case. The detailed numbers of patients with endocrinopathies are presented in Table 3. Among those who initially had visual impairment, 14 patients declared worsening of the symptoms, while the others did not report such symptoms until the postoperative time.

Toxicity was observed in only four patients after RTH was applied as a part of the primary treatment, and in 14 (32%) in total. The most common were endocrinopathies (four cases), vascular complications (four cases), and visual deficits (three cases) (Table 3). In one patient, a secondary tumor within the irradiation field was diagnosed—high-grade glioma. The patient died due to the progression of that tumor.

### 3.6. Overall Survival

The median follow-up was 8.4 years. During that time, six patients died, including one due to progression of craniopharyngioma and five due to other reasons or the reason of death was unknown. Five- and ten-year OS were 98% and 93%, respectively. The presence of endocrine deficits at the time of the diagnosis had a negative impact on survival in the univariate analysis, but this effect was not confirmed in the multivariate analysis (*p* = 0.055, HR 0.95, Table 4).

## 4. Discussion

Due to their location in the parasellar region, the diagnosis of craniopharyngiomas is often preceded by the occurrence of symptoms related to their local growth [1]. According to the current literature, around 40 to over 70% of patients present at least one endocrine deficit at the time of diagnosis, with much more after the surgery [1,3,9,11,13,14,49,50,51,52]. The initial 29% rate of endocrinopathies observed in our group is in accordance with those reports. Also, as high as 90% rate of postsurgical endocrine deficits was observed in studies from United States, Canada, Europe, and Asia [1,49,50,51,52,53,54,55,56,57,58,59,60,61,62,63]. The most commonly observed deficits concern the thyroid– and adrenal–pituitary axis, and this is also in accordance with the data from other countries [49,50,51,52,53,54,55,56,57,58,59]. In Yaxian et al.’s study in Chinese patients, being a prepubertal girl was found to be risk factor for impaired pituitary–thyroid and pituitary–adrenal axis function [52]. Due to the long study period (and changes in the Polish medical system), some patients, especially in the early years of the study, could have delayed endocrine assessment until the postsurgical time, which could also affect the estimations of the toxicity of surgery. In reports of other authors, 55–84% of patients reported visual impairment, and it was also commonly present at diagnosis in 46% of cases in our group [1,3,9,11,52]. The numbers vary due to the position of the chiasm in relationship to the tumor (anterior vs. posterior) as well as asymmetric extension of the tumor [1]. Other common symptoms present in our patients and in the literature include: headaches, neurologic deficits, and weight gain [1,11,49,50,51,52,53,54].

Neurosurgery is the primary treatment in the vast majority of patients with craniopharyngioma. As very good survival is observed in over 90% of patients with this low-grade tumor, the goal is to avoid the long-term morbidity [1,3]. A preoperative radiological grading system evaluating hypothalamic involvement has been developed for pediatric patients with the aim to optimize treatment strategy [1,3,11,32,64,65,66,67,68]. Several classification systems based on surgeon experience have been proposed and commented by Magill et al. [64]. The first classification was created in 1990 by Gazi Yaşargil and based on the relationship of the tumor to the surrounding structures (diaphragma sellae, the tuber cinereum/floor of the third ventricle, and the third ventricle/hypothalamus) [65]. Kassam et al. created their classification scheme for the purpose of endonasal approach for resection of craniopharyngiomas, which was further developed by Jamshidi et al. [66,67]. The last classification system based on presumed tumor origin was created by Fan et al. [68]. They classify tumors as “QST” types: infrasellar/subdiaphragmatic (type Q-CP); subarachnoidal (type S-CP); and pars tuberalis/third ventricle (type T-CP). Their results showed that the visual outcomes were significantly better in the endonasal group compared to the transcranial group, while in type T-CP, improved hypothalamic status was observed in the transcranial group [68].

During the analyzed period, the strategy for dealing with craniopharyngiomas has changed. Initially, the goal of the surgery was to achieve maximal tumor resection, aiming for complete removal whenever possible. Since 2008, our approach has become more conservative due to the demonstrated correlation between aggressive resection and the development of significant endocrinological deficits. Currently, the treatment strategy involves complete resection if the tumor does not involve the hypothalamus (type 0 or 1 according to the Puget classification), or partial resection with preservation of hypothalamic structures if the tumor involves this area (type 2 according to Puget) followed by radiotherapy. This change in strategy was found to be associated with a reduction in adverse endocrinological effects [1,5,32,62]. Transcranial approaches are traditionally used in the surgical treatment of craniopharyngiomas. However, recently, the endoscopic transnasal approach has been used more and more often [64,65,66,67,68]. Both approaches have their advantages and disadvantages. Due to direct access to the parasellar compartments, transcranial approaches are more useful for tumors that extend laterally [33,62,68]. Endoscopic transnasal procedures provide direct access to the anterior skull base and make the best choice for intrasellar lesions [34,68]. Some recent studies indicated that the endoscopic transnasal approach outperforms the transcranial one in terms of gross total resection rate and visual outcomes, as well as fewer endocrinological complications such as diabetes insipidus and panhypopituitarism [1,33,35,36,37,38,39,40,62,63,64,65,66,67,68]. However, the choice of approach should be individualized for every patient, depending on the tumor anatomy [36,64,65,66,67,68]. In most cases, we used a lateral subfrontal approach through a pterional craniotomy. The interhemispheric transcallosal approach was preferred when the tumor extended high into the third ventricle, obstructing the Monro foramina and hydrocephalus. Some patients underwent a two-stage operation—in the first stage, partial resection of the tumor was performed in the parasellar and lower-anterior third ventricle region using the lateral subfrontal approach and translamina terminalis. Then in the second stage, resection of the tumor in the upper part of the third ventricle was performed through an interhemispheric, transcallosal approach. As mentioned above, an endoscopic, transnasal approach is increasingly preferred in contemporary practice [1,62,64,65,66,67,68]. A staged surgery approach was evaluated by an Italian group, and Agresta et el. found that comparable outcomes in terms of clinical and oncological outcomes in single-stage and staged surgery could be observed, however more endocrine deficits were present in the staged surgery group [55]. A systematic review conducted by Clark et al. showed that comparable outcomes in terms of PFS could be observed in patients treated with gross total resection alone and in those who had subtotal resection followed by postoperative irradiation [6]. Thus, instead of maximally radical surgery, partial or subtotal resection without hypothalamic damage was advised [1,3,5,11,14,32,62]. Similar results were observed in several retrospective series as well as in one prospective German trial and in recent study from the RT 1 protocol and confirmed that non-radical surgery with adjuvant irradiation could provide very good outcomes in terms of progression-free survival and overall survival [1,5,30,31,32,32,33,33,34,35,36,37,38,39,40,41,42,43,44,57,62]. Also, a recently published study from a German group (RT2CR) confirmed these results [58]. The results observed in our group are in accordance with the studies of other authors and showed that the best outcome could be obtained with gross total resection or with combination of less radical surgery with adjuvant irradiation. What is more, a recent study by Aldave et al. showed that comparable results in terms of intellectual function and quality of life could be observed in those who had gross total resection and in those who had less radical surgery with adjuvant radiotherapy (both proton and photon radiotherapy techniques were used) [57]. Merchant et al. in their recent trial reported that cognitive outcomes with proton therapy were improved over photon therapy [58].

The total dose used in standard conventional RTH is around 54 Gy, delivered in 30 fractions over a period of six weeks. In their literature review, Yang et al. evaluated the outcome of 442 patients who underwent total (58%) or subtotal (23%) resection combined with RTH (in 19%). The PFS and OS were compared and showed inferior outcomes in the subtotal resection group; however, patients who had adjuvant RTH did not differ from those who had total resection [2]. This is in accordance with the data from the trial KRANIOPHARYNGEOM 2000/2007, which indicated that patients after incomplete resection not followed by RTH experienced progression more commonly than those who received adjuvant irradiation [5]. A recently published prospective study from the RT1 protocol and RT2CR has confirmed the beneficial role of immediate RTH [42,58]. Our study also demonstrated that patients benefitted from postoperative irradiation, both in the primary setting when RTH followed surgery or in the recurrence treatment when RTH was the only treatment applied or when it was combined with surgery. In all of them, RTH was well tolerated, and the rate of complications was similar to studies by other authors and was comparable between proton and photon RTH [1,9,47,53,57,58].

Another approach is radiosurgery with total doses between 5 and 15 Gy, similarly to doses used in the case of patients treated in our study [47]. The results published by Pikis et al. and in their literature review described 5-year PFS of 42 to 90% with excellent OS. Since no optimal timing or dose was associated with better outcomes, this approach is reserved for patients with selected small tumors after careful postoperative examination [47]. Additionally, several patients in our group received either adjuvant stereotactic RTH alone or stereotactic boost after the end of conventional irradiation. Radiosurgery was applied both in primary and in salvage treatment.

An attempt to avoid morbidity associated with surgery was described in the study of Young et al.—their patients had RTH alone without previous surgery [45]. Results from that study were excellent in terms of local control (one tumor progression 8.5 years after the treatment) and overall survival (all patients alive). Furthermore, there were no significant changes in vision, hearing, or neurologic function due to RTH. The major limitation of this study is the lack of second histopathologic confirmation of craniopharyngioma. Hill et al. and Drapeau W. et al. performed the systematic literature review that included retrospective studies that used definitive RTH, which had comparable results to gross total resection and subtotal resection combined with adjuvant irradiation [1,41]. A similar approach is proposed by Buchfelder M. et al. [62]. In our group, one patient received radiotherapy alone and has been diagnosed with tumor progression after 33.3 years. Based on those observations, it appears that RTH alone could be considered when radical resection is contraindicated. However, such a decision should be carefully made by the interdisciplinary team. Treatment morbidity is high because the tumor is located close to important surrounding structures [1,3,11,43,46,47,48,49,50,51,52,53,54,62]. Some authors observed that patients treated with radical surgery who experience tumor progression and receive RTH experience the greatest risk of complication, while others suggested that different treatment approaches could lead to similar rates of late complication if the follow-up period is extended [1,45,46,47,48,49,50,51]. Owing to the excellent long-term OS, minimizing the risk of toxicity from RTH and morbidity from surgical intervention is paramount [1,11,43]. Factors that were found to impact the progression-free survival are hypothalamic involvement, shunt at diagnosis, younger age, and tumor size [32,56].

Some authors reported visual impairment related to the surgery in 55–84% of patients, which was also confirmed in our group with 49% of patients with more pronounced/new visual deficits after the surgery [1,3,9,11,42,48]. Among those who initially had visual impairment, 14 patients declared worsening of the symptoms, while the others did not declare such symptoms until the postoperative time. Visual deficits after RTH were observed in 10% of patients who received proton therapy in the study of Jimenez et al., in four in the RT2CR trial, and in three cases in our group [9,58]. Several other authors reported comparable rates of visual impairment after radiotherapy of approximately 14% [12,46]. The 91% rate of endocrinopathies observed in our group is quite high; however, it could be the result of a very long and detailed follow-up in. Furthermore, cooperation with the endocrine department led to a very meticulous evaluation of deficits regarding their onset. In studies with a very long observation period, similar rates of endocrine deficits were noticed [1,11,12,48,49,50,51,52,53,54]. The worsening of already existing endocrinopathies were reported in all the studies in which radiotherapy was used, however, the rate of such complications varies between the studies—from 7% to 47% [3,9,44,45,50,51,52,53]. When surgery is combined with radiotherapy, some complications could be observed many years after the treatment, and it is difficult to precisely define the reason for hypothalamic–pituitary axis damage. Nevertheless, recent studies have indicated that the number of patients with endocrinopathies is lower when subtotal resection is combined with irradiation in the primary setting [50,51]. We were unable to conduct such analysis due to the lack of uniformity in the management of patients in our group.

Radiation therapy targeting the sellar/suprasellar region will unfortunately also include the intracranial carotid arteries and the circle of Willis. Authors reported up to 20% of cerebrovascular events after RTH, and this group is regarded as a particularly vulnerable group for vascular complications [42,46,58]. Furthermore, in our group, three patients experienced vascular complications after the surgery and four more experienced vascular complications after the RTH. The number of studies that reported the presence of tumors related to RTH is small, but some authors have found that the use of radiotherapy, especially at younger age, was associated with a higher risk of developing secondary brain tumors [48]. However, this hypothesis was not confirmed by others and in the majority of studies with follow-up exceeding 10 years, no secondary tumors were observed [12]. In their literature review of over 600 patients, Kiehna et al. found only four secondary tumors, all of which were high-grade gliomas. This is similar to the case observed in our group [42]. Furthermore, in the study of Edmonston et al., one case of high-grade glioma was diagnosed 13 years after RTH [42]. Young et al. reported one case of out-of-field malignancy [45]. However, the diagnosis of osteosarcoma one year after the irradiation within the field covered with a dose of less than 3 Gy suggests a lack of the correlation with previous irradiation, since such radiation-associated sarcomas are usually diagnosed after a median time of 12 years after primary treatment and are located in the region covered with a higher dose (over 30 to 40 Gy). With advances in photon irradiation and the implementation of proton RTH, the risk is relatively small; however, it is not negligible due to excellent treatment outcome and very long follow-up time in the case of pediatric patients. The follow-up should include careful observation with MR imaging in all craniopharyngioma survivors [1,60].

The management of craniopharyngiomas in children and young adults should start from the surgery [1,7,59,62]. Based on the presurgical MRI assessment, adequate type of surgery (partial/total/ biopsy or other) should be discussed with the patient and his/her parents depending on the age of diagnosis. Surgery should be performed at a neurosurgical department with experience in pediatric craniopharyngioma. The goal is to avoid a high level of postsurgical morbidity. The current Children’s Cancer Leukemia Group (CCLG) guidelines for craniopharyngioma suggested that in case of large tumors with hypothalamic involvement based on Paris grading, limited surgery should be followed by upfront radiotherapy, and radical resection should be reserved for smaller tumors without hypothalamic involvement. Radiotherapy is advocated in all cases after incomplete resection. Total dose of 54 Gy administered over 6 weeks could be given with photons or protons. Close, long follow-up with MR imaging is necessary [7,61].

There are several limitations of our study, which are typical for such retrospective analyses. The very long time period over which patients were treated with different treatment sequences and modalities (which changed over the last 40 years) are the main drawbacks. Despite the very long time, the number of patients included in the study is still relatively small, however comparable to the number observed in many reports. Due to the rarity of this tumor and only one prospective trial, studies like ours are a valuable way to collect the experience on this topic and source of the data for meta-analysis [5]. Another limitation is lack of the review of the pathologic diagnosis and no molecular analysis of pathologic specimens, but we were unable to collect the tissue samples for analysis. Evaluation of surgical outcome was not performed with unified imaging modality due to the long study period and lack of guidelines in the early years of the study period. However, based on the available data, the extent of the resection was evaluated according to the information provided in the patients’ charts by the experienced neurosurgeon. The timing and type of RTH varied among patients, and some were irradiated with old techniques. Nevertheless, the multidisciplinary approach with meticulous endocrine deficit’s evaluation, very long follow-up period, and the number of collected information regarding the treatment and diagnosis applied are factors which strengthened our findings. Lack of the evaluation of quality of life and neurocognitive function are also limitations of our study; however, such analysis is not possible in a retrospective study. The consensus regarding the management of children with craniopharyngioma for a very long time has evolved based on the institutional experience and retrospective studies like ours. We believe that multi-institutional or cooperative group prospective studies or registries are needed to better define these populations and inform future clinical investigations.

## 5. Conclusions

The treatment outcome of children with craniopharyngioma is good, and surgery combined with radiotherapy provides comparable results to gross tumor resection in terms of progression-free survival in the majority of patients. The rate of endocrinopathies in craniopharyngioma survivors is high. Adjuvant irradiation, applied in primary or recurrence treatment, improves disease control. Children who have been treated for craniopharyngioma require close follow-up with radiological and endocrinological assessment.

## Figures and Tables

**Figure 1 children-10-01272-f001:**
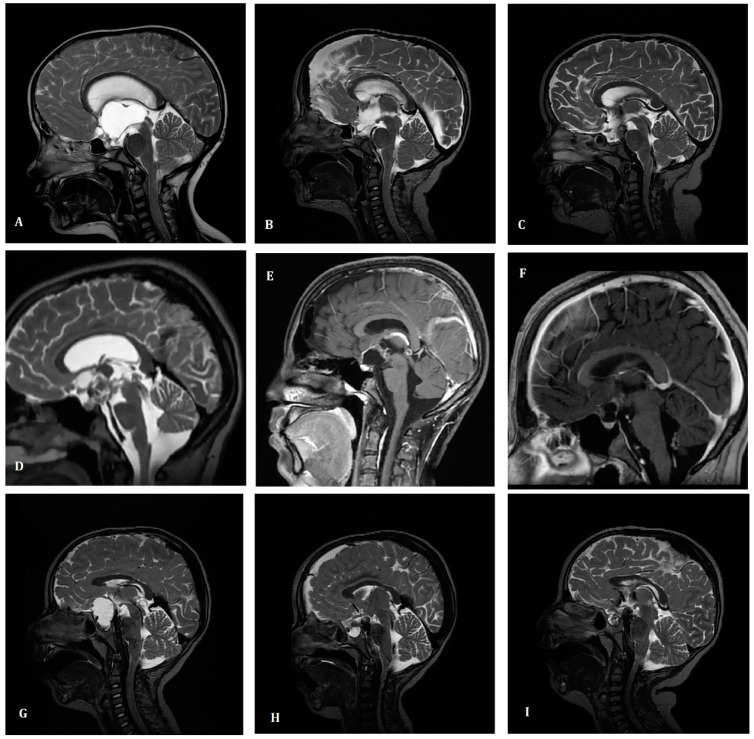
Examples of magnetic resonance imaging of treated patients. Case 1 (upper row) Six-year old boy treated with partial surgery. Conventionally fractionated irradiation was applied due to the progression of the size of the tumour. Magnetic resonance imaging (MRI) performed before the surgery (**A**), 3 days after the surgery (**B**) and six months later (**C**). Initial tumour dimensions were 56 × 49 × 39 mm and patient suffered from preoperative visual deficits. After the surgery endocrine deficits were diagnosed. After the irradiation the size of tumour is stable and no complications were present. Currently he is alive in follow-up and increase in body mass index is observed. Case 2 (middle row) Fourteen-year old boy treated with partial surgery and postoperative conventionally fractionated irradiation. Magnetic resonance imaging (MRI) performed before the surgery (**D**), 2 days after the surgery (**E**) and three and half months later (**F**). Initial tumour dimensions were 35 × 30 × 34 mm and patient suffered from preoperative headaches, visual deficits and increase in weight. After the surgery endocrine deficits were diagnosed. After the irradiation the size of tumour is stable and no complications were present. Currently he is alive in follow-up. Case 3 (lower row) Four-year old boy treated with partial surgery. Single fraction (16 Gy) of stereotactic radiotherapy was applied due to the progression of the size of the tumour. Magnetic resonance imaging (MRI) performed before the surgery (**G**), 7 days after the surgery (**H**) and one year later (**I**). Initial tumour dimensions were 23 × 36 × 28 mm and patient suffered from preoperative visual deficits (blindness of left eye) and endocrine deficits. After the surgery endocrine deficits were diagnosed. After the irradiation the size of tumour is stable and no complications were present. Currently he is alive in follow-up.

**Figure 2 children-10-01272-f002:**
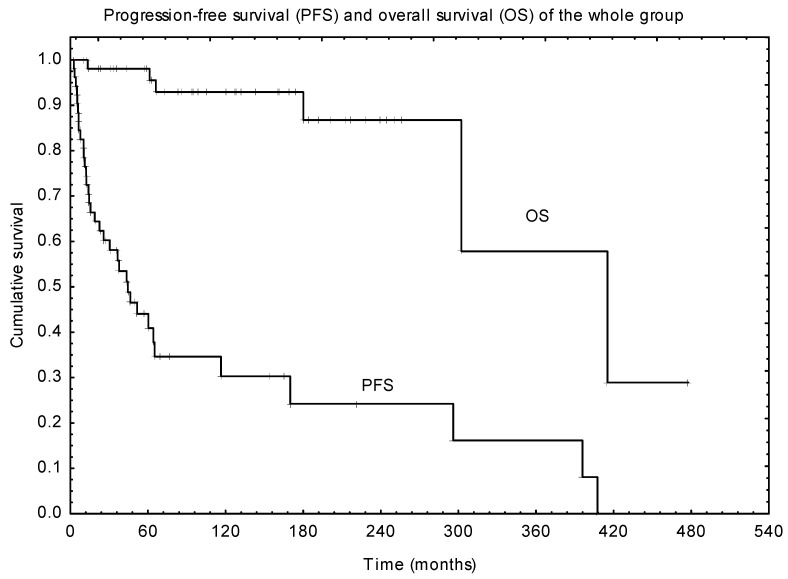
Progression-free survival and overall survival.

**Table 1 children-10-01272-t001:** The patients’ and primary treatment characteristics.

Characteristic		
		Value (range)
Age at diagnosis	Mean (range)	11 (2–21)
Tumor size at diagnosis	Median (mm)	30 × 30 × 30
	Range (mm)	7–93
		Number of patients (%)
Gender	Female	26 (46%)
	Male	30 (54%)
Underwent surgery in primary treatment	Yes	55 (98%)
	No	1 (2%)
The extent of primary surgery	R0	7 (13%)
	R1	31 (55%)
	R2	12 (22%)
	R3	2 (4%)
	R4	3 (6%)
Underwent RTH in primary treatment	Yes	18 (41%)
	No	26 (59%)
Type of primary treatment	R0 surgery only	7 (13%)
	Surgery + RTH	18 (32%)
	R1–R4 surgery only	31 (55%)
Follow-up	Median (years)	8.4
	Range (years)	0.4–39.8
Local failure or death	Yes	41 (73%)
	No	15 (27%)

R0—total radical resection, R1—subtotal resection, R2—partial resection, R3—tumor biopsy, R4—cyst drainage; RTH—radiotherapy.

**Table 2 children-10-01272-t002:** Prognostic factor for progression-free survival: univariate and multivariate analysis.

Univariate Analysis				
Variable		HR	95% CI	*p*-value
Age (years)		1.04	0.96–1.12	0.337
Sex	female	Ref.		
	male	0.95	0.48–1.87	0.881
Preoperative visual deficits	Present	2.36	1.14–4.87	0.021
	Absent	Ref.		
Type of surgery	Radical resection	0.39	0.12–1.30	0.125
	Non-radical resection	Ref.		
Radiotherapy in primary treatment	Yes	0.25	0.09–0.67	0.006
	No	Ref.		
**Multivariate analysis**				
Preoperative visual deficits		1.89	0.92–3.90	0.084
**Radiotherapy in primary treatment**		0.24	0.09–0.64	**0.005**
Type of surgery		0.32	0.10–1.06	0.063

**Table 3 children-10-01272-t003:** The evaluation of the endocrinopathies before and after the treatment.

Type of Hormonal Deficiency	Number of Patients with Hormonal Deficiency at the Time of Diagnosis	Number of Patients with Hormonal Deficiency after the Surgery	Absolute Difference %	Number of Patients with Hormonal Deficiency after the Radiotherapy	Absolute Difference %	Total Number of Patients with Hormonal Deficiency
**GH deficiency**	7 (13%)	13 (23%)	+10%	3 (7%)	+4%	15 (27%)
**LH/FSH deficiency**	6 (11%)	17 (34%)	+23%	3 (7%)	+2%	18 (32%)
**ADH deficiency**	5 (9%)	35 (63%)	+54%	1 (2%)	0%	35 (63%)
**ACTH deficiency**	-	32 (57%)	+57%	3 (7%)	+2%	33 (59%)
**TSH deficiency**	1 (2%)	33 (59%)	+57%	2 (4%)	+2%	34 (61%)
**PRL deficiency**	-	1 (2%)	+2%	0%	0%	1 (2%)
**No data**	5 (9%)	7 (13%)	-	1 (2%)	-	-

Abbreviations: ACTH—adrenocorticotropic hormone, ADH—antidiuretic hormone, GH—growth hormone, LH/FSH—luteinizing hormone/follicle-stimulating hormone, TSH—thyroid stimulating hormone. The number in one column could exceed 100% as a single patient could have more than one hormonal deficiency.

**Table 4 children-10-01272-t004:** Prognostic factor for overall survival: univariate analysis.

Univariate Analysis				
Variable		HR	95% CI	*p*-value
Age (years)		1.14	0.91–1.43	0.243
Sex	female	Ref.		
	male	4.39	0.51–37.86	0.179
Preoperative endocrine deficits	Present	10.44	0.95–114.56	0.055
	Absent	Ref.		

## Data Availability

Please contact author for data requests. The datasets generated during and/or analyzed during the current study are available from the corresponding author on reasonable request.

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
