# Peer review of "The Price of Success—The Long-Term Outcomes of Children with Craniopharyngioma—Two Institutions’ Experience"

_children, 2023, doi:10.3390/children10071272_

Round 1

Reviewer 1 Report

 The authors investigated the long-term outcomes of the patients treated for craniopharyngioma. They successfully showed the long-term prognosis of patients and described the therapy effects. However, the manuscript suffers from numerous flaws as indicated below need to be addressed by authors to improve the quality of the article.

1. They described the meaningful data and results of analysis only in the text. It makes very difficult for their manuscript to be understand. They should modify the Tables and Figures and clearly show the data of patients’ characteristics and statistical analysis in Tables.

2. In order to make it easy understanding the characteristics of this cohort, they should show a table of whole patients’ characteristics in this study, including tumor size, type of treatment, disease recurrence or progression, and follow-up period.

3. They should show the data of univariate and multivariate analyses for the recurrence and survival in a table.

4. The meaning of Table 1 was unclear. They should describe the reason why they divided the patient cohort into three groups and state their interpretation of the difference between the three groups.

5. Although the Table 1 showed primary treatment characteristics, they failed to present the type of primary surgery or treatment in the Table.

6. In Discussion section, they often discussed using the previous data of their group and not using the data presented in this study. They should clearly discuss about the results of this study.

7. They should plot the censored data in Figure 1.

8. The first paragraph of Discussion section begins abruptly. They should modify it so that the paragraph can be understand by reading it alone.

All points should be included in the manuscript.

Reviewer 2 Report

SUMMARY OF THE STUDY

The price of success – the long-term outcomes of children with craniopharyngioma – two institutions experience

This paper is highly valuable as it aims to evaluate the long-term outcomes and treatment side effects of children and young adults (under 12 years old) who were treated for craniopharyngioma over the past 40 years (1979-2022) at two hospitals in Poland. The study found that the most common neurosurgical complications observed were visual and endocrine deficits, affecting 49% and 91% of patients, respectively. Additionally, 32% of cases experienced complications after radiotherapy. The median follow-up period was 8.4 years (ranging from 0.4 to 39.8 years), during which 27 patients passed away. The 5- and 10-year overall survival rates were 97% and 93%, respectively.

Based on the study's findings, the authors concluded that the treatment outcomes for children with craniopharyngioma are generally favorable. Combining surgery with radiotherapy yields comparable results to complete tumor removal in terms of progression-free survival for most patients. However, the incidence of endocrine disorders among craniopharyngioma survivors is high. The use of adjuvant radiation, either in initial treatment or for recurrent cases, improves disease control. It is crucial for children who have undergone treatment for craniopharyngioma to receive regular follow-up examinations, including radiological and endocrinological assessments.

This study holds both social and clinical significance, and I believe it aligns with the objectives of the Children's Journal. Overall, the study is interesting, although there are certain limitations that adversely affect its quality.

GENERAL COMMENTS

·         The title properly reflects the subject of the paper.

·         The abstract provides an accessible summary of the manuscript.

·         The keywords accurately reflect the content.

·         The introduction properly sets out the argument.

·         The results are clearly explained.

·         The conclusions also consider the limits of the study.

·         The methods are appropriate, and the results are clearly presented.

·         The paper has an appropriate length.

·         References are balanced, updated, and complete.

·         The figure and tables are clearly organized and described.

I agree with the authors: as a retrospective analysis, this study has several limitations. The main drawback is the extensive-time period over which patients were treated with different treatment sequences and modalities, which have evolved over 40 years. Despite the long duration, the number of patients included in the study is relatively small. However, as stated by the authors, it is comparable to many other reports, given the rarity of this tumor and the limited number of prospective trials. An important aspect to be considered is that the management consensus for children with craniopharyngioma has evolved over a long time based on institutional experience and retrospective studies. Therefore, differences in the results are not too surprising.

Another piece of information missing is the lack of evaluation of the quality of life and neurocognitive function and the lack of a review of the pathological diagnosis and the absence of molecular analysis of the pathological patterns.

Finally, the timing and type of radiotherapy varied among patients, and some were treated with outdated techniques. Nonetheless, our findings were strengthened by the multidisciplinary approach, meticulous evaluation of endocrine deficits, long follow-up period, and comprehensive information collected regarding treatment and diagnosis.

To address the limitations of the study and reinforce its findings, I would recommend expanding the discussion and introduction sections with information from other international studies. This approach would allow the authors to contextualize their findings within a broader research landscape.

By examining studies conducted in different territories, the authors can identify the influence of regional variations in risk factors and their impact on the prevalence, diagnosis, and treatment of cancer in children and young adolescents.

Moreover, by considering the concordance of results across various countries, it is possible to identify commonalities and differences in risk factors, which can provide a more comprehensive understanding of the topic.

Additionally, I suggest including guidelines to guide new approaches to managing craniopharyngioma in children and young adolescents. Guidelines can help healthcare professionals standardize their practices and ensure that patients receive the best possible care. Moreover, the inclusion of guidelines in the study can provide practical recommendations based on the findings, helping clinicians develop evidence-based strategies for diagnosis, treatment, and follow-up care for this particular population.

In conclusion, by incorporating an international perspective, referencing relevant studies, and providing guidelines, the authors can strengthen the study's impact and contribute to the advancement of knowledge and care in the field of pediatric oncology.

English typos and repetitions should be corrected.

English style should be revisioned by an English mother tongue.

Reviewer 3 Report

Authors present a retrospective review on 56 patients average age of 11 years which were treated at single center for past 40 years on craniopharyngeoma. 55 patients underwent surgery, in only 29 single surgery was performed. Mean follow up of 8 years was noted, five year  progression free survival for R0 resection, resection with adjuvant RTH and R1-R4 resection was 29 83%, 68% and 23%, respectively; complication rate was 32%.  Surgery combined with radiotherapy provides comparable results to gross tumour resection in terms of oncologic outcome in craniopharyngioma patients.

Low number of patients and retrospective character of the study are its main drawbacks, as well as various surgeons. I suggest to present an evolution of the surgical technique, i.e. the improvement of results - survival, outcomes, complications, over the course of 30 years. I also suggest to put all the patients in a Table. There is no single illustrative case - please include at least three with pre and postoperative MRI and long term follow up. What was the rate of adiposity as a very common sideffect in pediatric patients following this surgery? Please define R0 and R1 resection - this term is unusual in brain tumor surgery. How many patients did get their pituitary stalk resected? What was the rate of hydrocephalus, how many patients had VP shunt, cysts, cyst-shunts? 

I suggest to broaden the discussion with surgical aspects -why was which approach chosen for a certain tumor, what was behind it, how did the morphology and location dictate the approach choice. 

For discussion please include and comment following papers: 

Drapeau A, Walz PC, Eide JG, Rugino AJ, Shaikhouni A, Mohyeldin A, Carrau RL, Prevedello DM. Pediatric craniopharyngioma. Childs Nerv Syst. 2019 Nov;35(11):2133-2145. doi: 10.1007/s00381-019-04300-2. Epub 2019 Aug 5. PMID: 31385085.

Buchfelder M, Schlaffer SM, Lin F, Kleindienst A. Surgery for craniopharyngioma. Pituitary. 2013 Mar;16(1):18-25. doi: 10.1007/s11102-012-0414-8. PMID: 22836237.

Ezzat S, Kamal M, El-Khateeb N, El-Beltagy M, Taha H, Refaat A, Awad M, Abouelnaga S, Zaghloul MS. Pediatric brain tumors in a low/middle income country: does it differ from that in developed world? J Neurooncol. 2016 Jan;126(2):371-6. doi: 10.1007/s11060-015-1979-7. Epub 2015 Oct 29. PMID: 26514358.

Please provide classification according to Magill. 

Acceptable. 

Round 2

Reviewer 1 Report

 The authors successfully revised their manuscript. The Figures and Tables have been well presented and make the manuscript easy to understand. The discussion section was also well-reorganized. This study would make a useful contribution in the treatment and management of patients with craniopharyngioma.

Reviewer 2 Report

Now the paper has been significantly improved and it can be accepted for publication in Children

English has been improved

Reviewer 3 Report

Authors have sufficiently responded to reviewer remarks.

Acceptable.